# Multi-scale modeling of intensive macroalgae cultivation and marine nitrogen sequestration

Meiron Zollmann[1✉], Boris Rubinsky[2], Alexander Liberzon [3] & Alexander Golberg[1]

Multi-scale macroalgae growth models are required for the efficient design of sustainable, economically viable, and environmentally safe farms. Here, we develop a multi-scale model for *Ulva* sp. macroalgae growth and nitrogen sequestration in an intensive cultivation farm, regulated by temperature, light, and nutrients. The model incorporates a range of scales by incorporating spatial effects in two steps: light extinction at the reactor scale (1 m) and nutrient absorption at the farm scale (1 km). The model was validated on real data from an experimental reactor installed in the sea. Biomass production rates, chemical compositions, and nitrogen removal were simulated under different seasons, levels of dilution in the environment and water-exchange rate in the reactor. This multi-scale model provides an important tool for environmental authorities and seaweed farmers who desire to upscale to large bioremediation and/or macroalgae biomass production farms, thus promoting the marine sustainable development and the macroalgae-based bioeconomy.

---

[1] Porter School of Environmental and Earth Sciences, Tel Aviv University, Tel Aviv, Israel. [2] Department of Mechanical Engineering, University of California at Berkeley, Berkeley, CA, US. [3] School of Mechanical Engineering, Tel Aviv University, Tel Aviv, Israel. ✉email: meironz@mail.tau.ac.il

Marine conservation and sustainable development is essential for achieving the United Nations' Sustainable Development Goals[1,2]. Large scale seaweed (macro-algae) farms (> 1 km) could proffer a sustainable and environmentally safe means for biomass production for biorefineries without expanding agricultural lands or freshwater requirements. Such macroalgal biorefineries[3–7] could supply the soaring demand for food, energy and raw materials. Furthermore, seaweed aquaculture can be utilized for eutrophication mitigation[8–11], thus contributing to the international effort to abate nutrient over-enrichment in coastal ecosystems[12,13] (i.e. the Mediterranean Action Plan[2]). However, the implementation of commercial cultivation of seaweed beyond East Asia countries is limited, because of a lack of farming tradition, undeveloped markets, and a questionable economic viability[14]. Large-scale commercial macroalgae cultivation, which is considered a new technology in most countries, could be advanced using multi-scale models. The use of multi-scale models to promote new technologies in reduced time and cost was demonstrated in the Carbon Capture Simulation Initiative[15]. The Carbon Capture Simulation Initiative, a partnership among national laboratories, industry, and universities, was established to enable accelerated commercialization of carbon capture technologies by developing multiscale models and simulation tools, used to improve design and reduce scale-up risk. Similarly, advances in cultivations of seaweed from small-scale activities to large scale implementation could also benefit from the availability of multiple scale models. We propose that these multi-scale models could facilitate the design and optimization of large seaweed farms by incorporating in the large scale models data from cultivation activities in a small scale[16,17], and demonstrate it in a study with mathematical and experimental parts.

Current macroalgae growth and nutrient dynamics models were developed for specific applications. For example, long-term ecological models that attempt to predict macroalgal productivity and seasonal blooms in prone ecosystems[18–29] or "black box" culture models that focus mostly on on-shore photobioreactors or tanks[11,30] or offshore extensive cultivation[31–33]. These models, which pursue a basic understanding of the thermodynamics of individual algae thalli and photobioreactors[34], can provide a general idea about productivity and seasonal effects on algae growth. However, they do not incorporate spatial effects at the scale of the farm and its environment and therefore cannot predict how the algae would behave in a real-life large-scale farm. On the other hand, as proposed above, multi-scale models that extend from the scale of a single plant to the scale of the farm could be used for the design of real-life scale seaweed farms[17]. Such a multi-scale model could incorporate available small-scale mathematical models and small-scale experimental data. This challenging task involves the combination of multiple biological, engineering and environmental factors and is the focus of this research.

Recently, some studies have proposed to apply intensified macroalgae cultivation, usually done in photobioreactors, also at near- and off-shore seaweed farms[35,36]. Intensified cultivation systems rely on frequent harvesting and could benefit from temporal multi-scale models that can predict biomass production and chemical composition in a time scale of days. As a case study, we used data from a Mediterranean Sea near-shore intensive growth experimental reactor used for free-floating *Ulva* species cultivation, which was described by Chemodanov et al.[36]. This reactor employs airlift pumps and bottom aeration and is suitable for shallow coastal areas or estuarine systems, in which macro-algae have a natural important role in nutrient cycling[26]. As these environments are the most prone to harmful eutrophication[37,38] which is responsible for significant environmental and economic

damages[38], the added value of nutrient bio-sequestration may increase the economic viability of seaweed cultivation in such locations.

Finally, the gap addressed in this study is the shortcoming of existing models in predicting biomass yield, biochemical composition, and ambient nitrogen concentrations in the farm scale. Thus, we develop a theoretical multi-scale model for macroalgae growth and nitrogen sequestration in an intensive cultivation seaweed farm, which is regulated by temperature, light and nutrients (Fig. 1). The model is used to simulate farm-scale biomass production and nitrogen removal in a nutrient-enriched environment, at a temporal and spatial resolution and scale that is not available today. Specifically, the model predicts farmed sea-weed biomass and sequestered N in different seasons. The model incorporates the required nutrient concentrations and how is the spatial distribution of biomass composition and productivity affected by levels of airlift pumping and dilution in the environment. Our model enables the investigation of farm spatial and temporal responses to environmental variations and provides useful insights on the effects of farm design and operation on the compliance with environmental and commercial requirements (i.e. uniform biomass composition and minimal energy consumption). Altogether, this multi-scale model provides an important tool for environmental authorities and seaweed farmers who desire to upscale to large bioremediation and/or macroalgae biomass production farms, thus promoting the macroalgae-based bioeconomy.

## Results and Discussion

**Calibrated model.** The calibration process, described in detail in the Supplementary Methods and results, started with light extinction parameters $K_a$ and $K_0$ and continued to growth function parameters (parameters of Eq. 1 and eq S1 in the Supplementary Methods). Based on a scan of 600 parametric combinations within a pre-defined range, which was built based on literature values (Supplementary Table 4), we manually fitted parametric combinations that provide both good RMSREs (<15%) and experiment-specific good relative errors (<20%). We used both criteria to prevent over- or under dominance of specific returns and environmental conditions (i.e. three returns with a low error and one with a high error). The chosen parametric combination yielded $RMSRE_1 = 10.3\%$ for the first step and $RMSRE_2 = 13.7\%$ for the second step (Supplementary Figures 6-9 and Supplementary Table 5), which are reasonable average relative errors for biological models[39,40]. Furthermore, these relative errors are significantly lower than the errors of models published in prior literature, relating to *Ulva* sp. growth in natural environments (i.e. 35–110%[41]), demonstrating the potential advancement in our dynamic model.

*Light extinction parameters.* We found that the model is not sensitive to $K_0$ in the examined range as the optical path in water is short. The best fit between in- and ex-situ light intensity measurements were found using a light extinction coefficient of $K_a = 0.15$ (Supplementary Figure 10), which is higher than the previously used $K_a = 0.01$[11] for *Ulva*, but similar to values used for other algae species[42]. The higher value better represents the significant effect of biomass density on light extinction.

*Growth function parameters.* $f_{Temp}$ parameters, $T_{opt}$ and $T_{max}$, were adjusted to 18 and 31.5 °C, fitting the literature optimal temperature range of 15–20 °C[43,44]. $K_I$ was adjusted to 20 μmol photons m$^{-2}$ s$^{-1}$ (Supplementary Figure 11). However, $K_I$ is a flexible parameter and is known to decrease when the *Ulva* is acclimated to low light intensities[45]. $\lambda_{20}$ was adjusted to 2.2% day$^{-1}$ (0.16% light hour$^{-1}$,

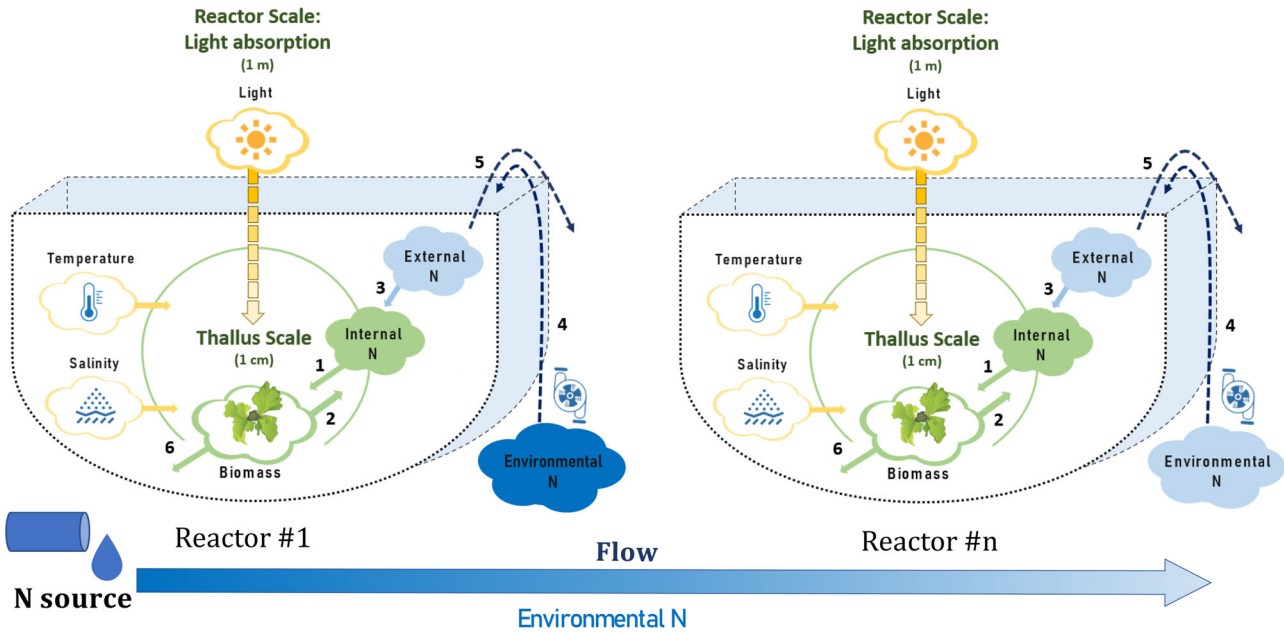

**Fig. 1 A schematic description of the multi-scale model.** The thallus scale (1 cm, green circle) is composed of a simple metabolic model of *Ulva*, in which the production of new biomass (*Ulva* icon) is affected by internal nitrogen (N, full green cloud) and by constraining environmental conditions, including light intensity, salinity and temperature (yellow clouds). The reactor scale (1 m, U shape pictures) adds light extinction effects (yellow graduated arrow), the concentration of external N in the reactor and the concentration of environmental N outside the reactor (dark/light blue clouds, depending on N concentration). The farm-scale (1 km, row of reactors starting at Reactor #1 and counting downstream to Reactor #n) adds the nutrient reduction caused by absorption in reactors along with the flow (Blue graduated arrow). Green and blue clouds represent the model state variables. Numbers represent the following processes: 1. Biomass growth; 2. Dilution of internal N by growth; 3. N uptake; 4-5. Water exchange by airlift pumping and overflow, and 6. Biomass losses.

Supplementary Fig. 11), which is low compared to literature values (5–6.5% day$^{-1}$). A limitation of this study is that the calibration system was mostly P-limited (N:P > 20[3]), a fact that is not represented in the model and may lead to underestimations of biomass production under P-saturation conditions. Furthermore, the agreement between modeled and measured final $N_{int}$ was low, which may be a result of the P-limitation, as high N:P ratio can inhibit N uptake[44].

**Sensitivity analysis.** The parameter with the largest total effect on the total biomass production and N bio-sequestration (Sobol sensitivity index of 0.35–0.4 in the range of 0 to 1) is $K_a$. $K_I$ and $\lambda_{20}$, with total sensitivity indexes of 0.15–0.28 and 0.09–0.1, respectively, have a moderate effect, and $\mu_{max}$ has a weak effect (~0.02) on total biomass production and N bio-sequestration. $N_{env}$, in comparison, is highly sensitive only to $d$ (sensitivity index of 0.97). The effect of other parameters within examined range is negligible (<0.01) (Fig. 2). This analysis shows that our multi-scale model is sensitive to parameters related to light ($f_I$), which, in the simulated climate, limits growth only in winter when days are short and sky may be cloudy, and when biomass density in reactor is high. The sensitivity of the model to parameters related to N ($\psi_{N_{ext}}$ and $f_{N_{int}}$), on the other hand, is low, as both reach a steady state relatively rapidly in N rich environments and affect model outcomes only when $N_{ext}$ and $N_{int}$ are low (i.e. $N_{ext}$ below $K_S$ or $N_{int}$ below $N_{intcrit}$). The low sensitivity to N related parameters can be understood in greater depth by the time-scale separation idea[46]. In diluting environments ($d > 0$), small changes in $d$ have significant effects on the results of the multi-scale model

as they force rapid $N_{env}$ and $N_{ext}$ attenuation regardless of biomass uptake. contrarily, small changes in $Q_P$ have no effect on model results as throughout the examined range N supply does not limit growth. The model was found to be insensitive to $f_{Temp}$ and $f_s$ related parameters in the simulated environmental conditions, but this finding should be examined with a wider range of temperatures and salinities. Model sensitivity to $\lambda_{20}$ was higher than the sensitivity to $\mu_{max}$ probably due to the dependence of $\mu$ also on other parameters (T, S, I and $N_{int}$), that lessen the direct effect of $\mu_{max}$ on model results.

**Seasonal trends in biomass production and nitrogen removal.** Productivity and N sequestration vary significantly seasonally, ranging between 0 and 26.8 gDW day$^{-1}$ m$^{-2}$ (0–30 gDW day$^{-1}$ m$^{-3}$) and between 0.2 and 1.2 gN day$^{-1}$ m$^{-2}$ (0.2–1.3 gN day$^{-1}$ m$^{-3}$), with average values of 13.3 gDW day$^{-1}$ m$^{-2}$ (14.9 gDW day$^{-1}$ m$^{-3}$) and 0.7 gN day$^{-1}$ m$^{-2}$ (0.8 gN day$^{-1}$ m$^{-3}$) (Fig. 3). In a farm of 100 chained reactors (cultivation area of 200 m$^2$), this translates into annual productivity of 1210 gC m$^{-2}$ year$^{-1}$, almost four times the estimated average productivity of terrestrial biomass in the Middle East (290 gC m$^{-2}$ year$^{-1}$ [47]) and N sequestration of 249 gN m$^{-2}$ year$^{-1}$. Peak production is expected from the end of February till the middle of March, and a second production peak is found in November. In contrast, production during the summer is very low. Level of N sequestration follow the same seasonal trend (Pearson's $r = 0.999$ in a non-diluting environment), although the correlation between production and N sequestration is expected to decrease in larger farms or in diluting environments, in which environmental N could be depleted (i.e. Pearson's $r = 0.996$ in the diluting

environment simulated below). The differences in N sequestration between the diluting environment ($d > 0$), in which high and low $N_{env}$ water is mixed, and the non-diluting environment ($d=0$) is discussed below in the spatial effects section. This seasonal variation follows changes in water temperature. Optimal water temperature (i.e. 18 °C, as found in the calibration of the model) lead to high productivities and high water temperatures, close to $T_{max}$ (i.e. 29.5 °C), lead to very low productivities (Supplementary Figure 5). Therefore, effective bio-

sequestration cannot be applied during the summer in the modeled conditions.

To reduce environmental N levels below a defined, environmentally benign, level, different seasons require different sizes of the seaweed farm. Considering that a 10 μM threshold prevents extreme eutrophication[48], to avoid damage to the environment, in winter, the dimension of the farm should be 1,462 m², in spring the farm should be 914 m² and in the fall 1,192 m² (Fig. 4a). From the perspective of the model, these dimensions of the farm are between 600 to 900 reactor size macro elements, i.e. the assumption that the single element control volume used in the analysis is small relative to the entire domain of analysis is acceptable. As important, these results demonstrate the value of this analysis. They provide a measure on how to design a large seaweed farm that is safe for the environment.

Following are additional examples of how this multi-scale model can be used to design large seaweed farms. A farm designed according to winter N sequestration abilities will produce 7.1 tons DW year⁻¹, whereas farms designed according to spring or autumn sequestration abilities will produce only 4.4 or 5.8 tons DW year⁻¹, respectively. As a general trend, in high $N_{env}$ levels, the relationship between added reactors and N sequestration is linear, but in lower N levels, closer to $K_S$, uptake is slower, and more reactors are needed per sequestration unit. Figure 4b–d present N and biomass dynamics in the last reactor in a farm designed to achieve the threshold in all seasons (731 reactors). Fixed year-round cultivation cycles result in time and space non-uniform chemical composition. Figure 4c also demonstrates how N content in the last reactor changes between seasons, with higher N content during winter and lower N content during autumn and spring, when productivity is higher. Uniform chemical composition, if required, can be achieved by adjusting lengths of cultivation cycles to environmental conditions, specifically, temperature, day length and $N_{env}$. Shortening autumn and spring cultivation cycles to 11 and nine days, respectively, for example, will enable the production of biomass with constant $N_{int}$, although won't comply with the defined 10 μM threshold during the spring (Supplementary Figure 12). However, shorter cultivation cycles come at an expense of higher labor demand and do not necessarily grant higher accumulated yields.

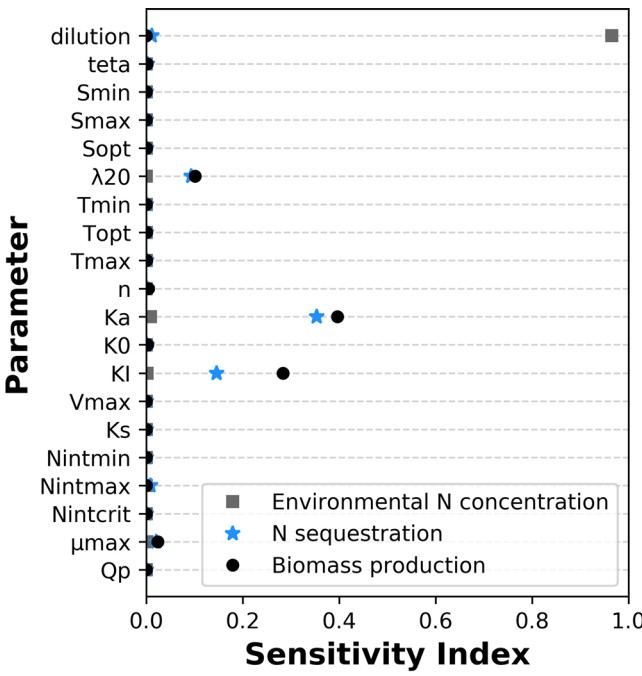

**Fig. 2 Illustrated sensitivity of simulated biomass production, N sequestration and final environmental N levels.** Simulated biomass production (black circles), N sequestration (blue stars) and final environmental N levels (gray squares) to model parameters, as measured by the Sobol method.

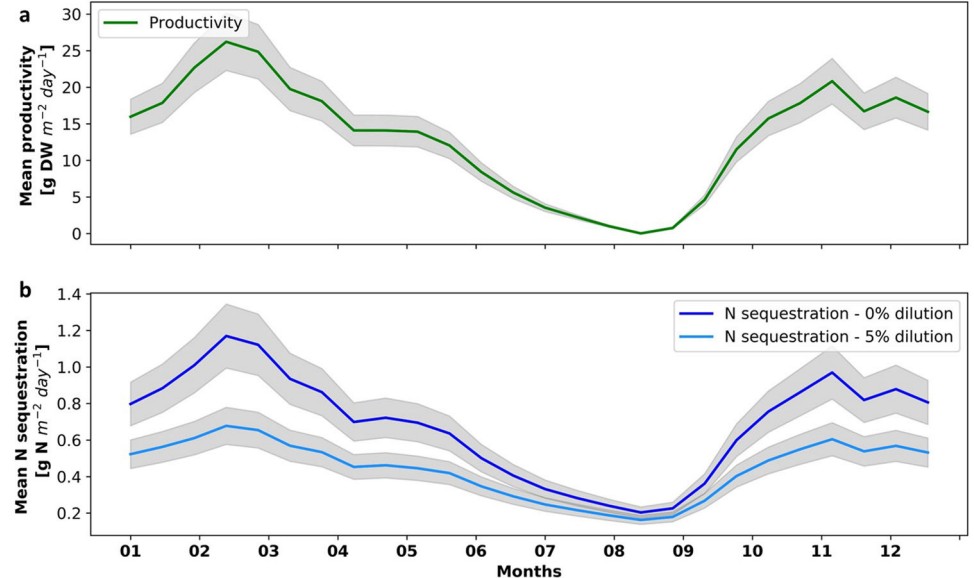

**Fig. 3 Year-round distribution of mean productivity and N sequestration in a seaweed farm, in two levels of simulated dilutions. a** Mean productivity (gDW m⁻² day⁻¹, green). **b** mean nitrogen sequestration (gN m⁻² day⁻¹) in a non-diluting environment ($d = 0$, dark blue) and a diluting environment ($d = 0.05$, 5% dilution between each two reactors, light blue) vs time of the year for a farm of 100 reactors. Shaded region represents a 15% calibration error.

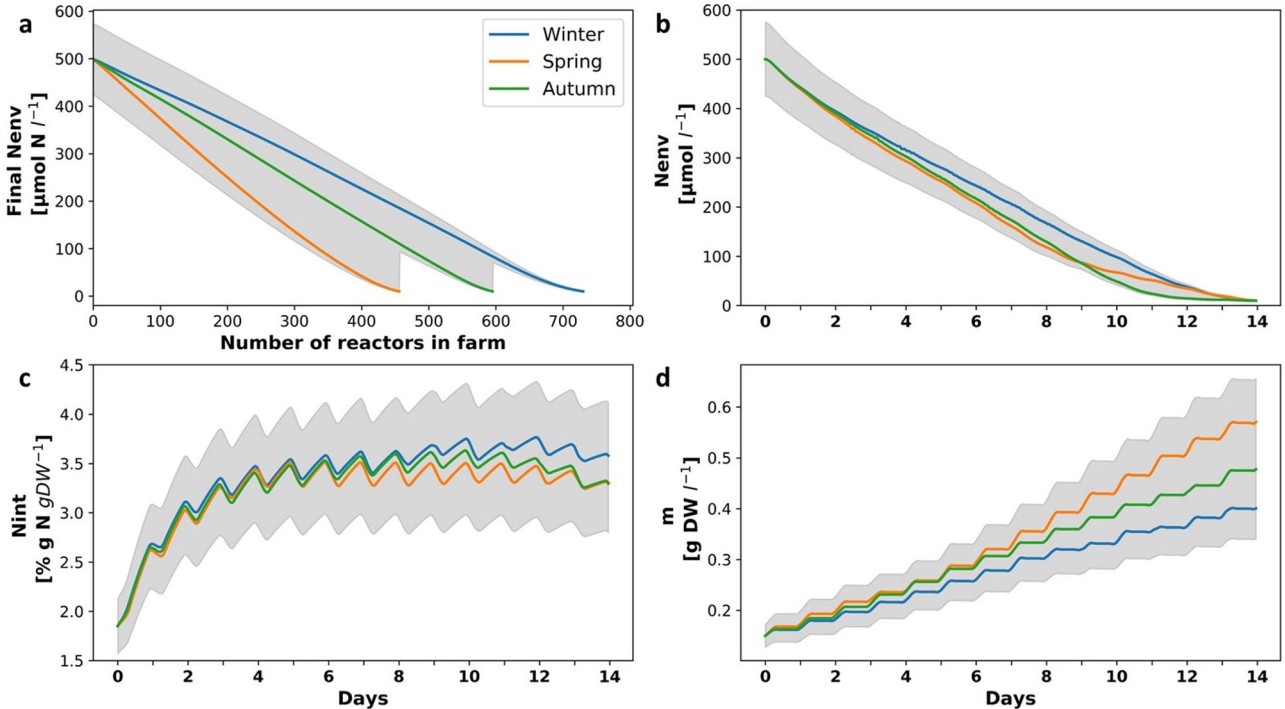

**Fig. 4 $N_{envr}$ $N_{int}$ and $m$ dynamics in a seaweed farm in different seasons, along a 14 days cultivation period. a** Final $N_{env}$ concentration (μM-N) after 14 days of cultivation as a function of the number of reactors in different seasons (Winter in blue, Spring in orange and Autumn in green). **b–d** $N_{env}$, $N_{int}$ and $m$ dynamics along 14 days' cultivation periods in different seasons, for the last reactor in a farm of 731 reactors. Shaded region represents a 15% calibration error.

**Spatial effects controlled by dilution and pumping.** In our model, spatial effects on biomass composition and growth rate appear only when $N_{env}$ decreases to limiting levels. The rate of this decrease can be controlled by airlift pumping flow and is accelerated in a diluting environment.

*Pumping flow.* $Q_P$ can be manipulated to control N flux into reactors and thus also chemical composition and growth rate of the algae (Fig. 5). The immediate effect of $Q_p$ is on the $N_{env}$ vs $N_{ext}$ dynamics. High $Q_p$ minimizes differences between $N_{env}$ and $N_{ext}$, which leads to a faster reduction in $N_{env}$ and slower reduction in $N_{ext}$ compared to the trajectories of $N_{env}$ and $N_{ext}$ with lower $Q_p$. Simulating reactors without pumps ($Q_p = 0$, dark blue line) decouples $N_{ext}$ from $N_{env}$ and eliminates the spatial effects of nutrient absorption. Thus, although $N_{env}$ does not change, rapid depletion of $N_{ext}$ leads to a decrease in $N_{int}$ which is followed by a decrease in produced biomass. Therefore, in the described system pumping is essential. High $Q_p$ promotes bio-sequestration but may result in a steeper spatial gradient of $N_{int}$ compared to low $Q_p$. Finally, $Q_p$ can be manipulated according to farm design requirements, controlling farm size and biomass composition. It should be mentioned that water exchange by pumping has additional important contributions, such as the supply of inorganic carbon, removal of waste material which may inhibit growth, and temperature control[36,49]. Furthermore, in an estuarine environment, pumping water from 1-2 m below the surface can increase salinity, which is crucial for the growth of marine macroalgae species. However, water pumping is an energy-consuming component of seaweed farms and should be optimized to minimize its carbon footprint. Previous trials to cultivate *Ulva* in the described reactors without water exchange were unsuccessful in our group[36]. However, a thorough review of seaweed cultivation[49] mentioned that water exchange in *Ulva*

cultivation can be reduced to 10% day$^{-1}$, equivalent to 15 l h$^{-1}$ in our work, without a significant change in yield.

*Dilution.* In highly diluting environments, bio-sequestration would be usually ineffective. However, such environments are not prone to eutrophication and do not require nutrient removal. Figure 6 presents the spring system dynamics in a 100-reactors farm, subjected to 5% dilution between each two reactors, similar to dilution rates used in literature[32]. Compared to the first reactor (darkest green), which is not affected by dilution, downstream reactors meet lower $N_{env}$ concentrations which are translated into lower $N_{ext}$ and gradually into lower $N_{int}$ and lower biomass production. In the simulated conditions ($N_{env0}$ = 500 μM), annual decrease in biomass production due to dilution (968 to 962 kgDW, 0.6%) is significantly smaller than the annual decrease in N sequestration (50 to 32 kgDW, 36%) (Fig. 3). This difference can be explained by the production of low protein biomass in the downstream, diluted, areas. Larger farms may not be practical in high-dilution locations, as downstream $N_{env}$ concentrations would not allow any growth beyond what the initial $N_{int}$ allows. However, using high-protein upstream biomass as a continuous seeding feedstock for further cultivation may enable sustainable low protein biomass production in such an environment. Following a similar concept, previous works suggested performing a two-step cultivation process, starting with high biomass production in a nutrient-rich environment and finishing with carbohydrate accumulation in nutrient-limited environment[50]. As opposed to the protein-rich biomass that is produced in N enriched environments and can be used for food and feed applications, such carbohydrate-rich biomass is advantageous for the extraction of different polysaccharides (i.e. starch, ulvan and cellulose) and can be processed into various forms of biofuels and chemicals[5].

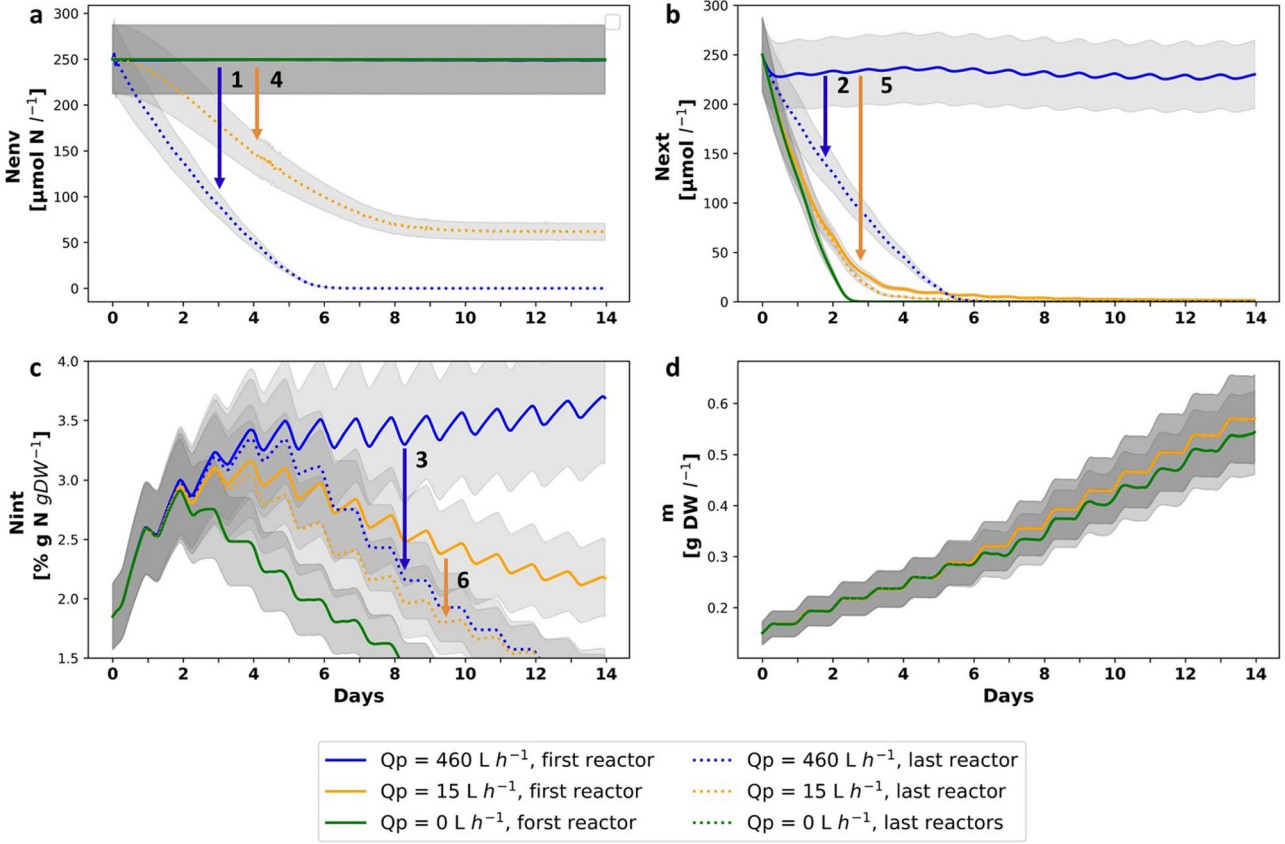

**Fig. 5 $N_{env}$, $N_{ext}$, $N_{int}$ and $m$ dynamics along a 14 days cultivation period, simulating $Q_p$ values of 0, 15, and 460 l h⁻¹.** Arrows highlight differences between first and last reactor: **a** $N_{env}$ dynamics: differences for $Q_p = 460$ l h⁻¹ and for $Q_p$=15 l hour⁻¹ are marked by (1) and (4), **b** $N_{ext}$ dynamics: differences for $Q_p$=460 l h⁻¹ and for $Q_p$=15 l hour⁻¹ are marked by (2) and (5), **c** $N_{int}$ dynamics: differences for $Q_p$= 460 l hour⁻¹ and for $Q_p$ = 15 l hour⁻¹ are marked by (3) and (6), and **d** $m$ dynamics. Simulation parameters and IC: 731 reactors, spring season, $N_{env0} = 250$ μM-N, $d = 0$. Shaded region represents a 15% calibration error.

A few previous studies assessed the effectiveness of eutrophication bioremediation in China by macroalgae cultivation. Generally, this was examined by comparing N and P open sea levels in cultivation season and off-season, by calculating how much nutrients were removed based on published data and biomass composition analysis, and by following eutrophication symptoms, such as hypoxia and harmful algal blooms[9,51,52]. One study, by Fan et al.[8], advanced into actively increasing nutrient removal by ecological engineering, specifically artificial upwelling, which is the pumping of nutrient-rich deep water to the surface. Fan et al.[8] found that artificial upwelling can increase the average yield of kelp seaweed by 55 g per plant, and developed a few useful recommendations regarding the conditions in which intensified cultivation can be worthwhile. Although in a different setup and framework, our work strengthens their recommendation to optimize pump operation according to algae requirements (nutrients, water exchange and salinity and temperature control), environmental conditions and regulations, and energy costs. These considerations change seasonally and spatially, even within the farm itself. Our model, developed especially for this cause, can help relating to spatial differences during the design and the operation of seaweed farms.

The environmental significance of this work relates to two major environmental issues: climate change and water pollution. The model developed in this work can be used to quantify and optimize the environmental significance of large-scale seaweed farms, specifically eutrophication mitigation. Thus, bioremediation by seaweed farms can be advanced from an unplanned external benefit to an inherent part of coastal development.

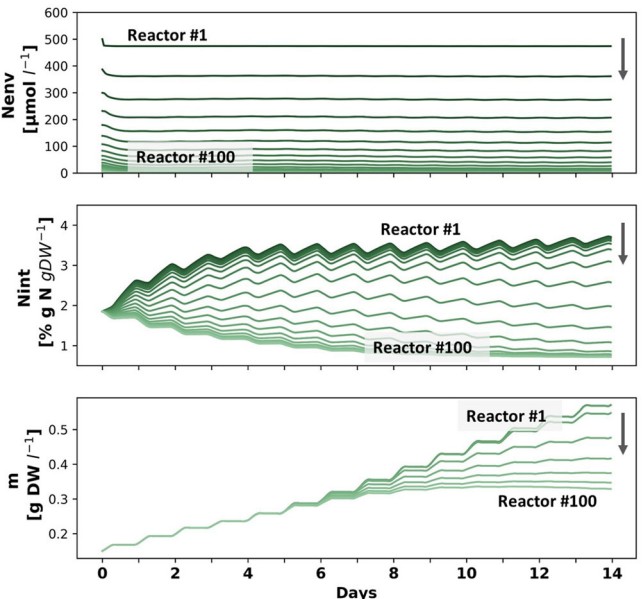

**Fig. 6 $N_{env}$, $N_{int}$ and $m$ dynamics along a 14 days cultivation period in a diluting environment in a farm of 100 chained reactors.** The lines represent $N_{env}$, $N_{int}$ or $m$ in each fifth reactor, starting from x=1 (darkest green) and progressing downstream along the arrows towards the last reactor (x=100, lightest green). Simulation parameters and IC: spring season, $N_{env0} = 500$ μM-N, $d = 0.05$.

**Fig. 7 Flowchart of study methodology.** Step 1: model formulation, including assumptions, governing equations and scale elements; step 2: model calibration, including light extimction and growth function parameters, and a sensitivity analysis, and step 3: model simulations, including seasonal trends in biomass production and nitrogen removal and spatial effects controlled by dilution and pumping.

Furthermore, if eutrophication mitigation is compensated by the authorities, this model can play a key role and incentivize the establishment of new seaweed farms, accompanied by additional environmental and economic benefits, on the local (i.e. marine conservation and economic development) and global (i.e. carbon sequestration, sustainable biomass supply and mitigation of fresh water stress) scales. In addition, with some modifications, this model can be used to model fish cages and integrated multi-trophic aquaculture (IMTA) and promote sustainable aquaculture and marine development. Altogether, this work utilizes model simulations to demonstrate production and sequestration potentials of macroalgae farms under different conditions and operation modes, and provides a quantitative tool enabling to promote the future deployment of such farms in large scales and maximize their benefits.

At the same time, this model is still theoretical as it was calibrated only in the reactor scale and using a small dataset. Farm scale calibration is not possible at this stage, as the modeled reactor is a pilot scale experimental reactor. However, alternative farm-scale calibrations may be performed in the future by adjusting the model to existing farm designs and calibrating it with field data. Additional limitations relate to the low-resolution data regarding nutrient concentrations in the water, intrinsic variations in biomass behavior which are currently not modeled and the lack of hydrodynamical data. Therefore, further research and development is required before this model could be useful for field application.

## Conclusions

We developed a multi-scale model for *Ulva* sp. macroalgae growth and nitrogen sequestration in an intensive cultivation farm, regulated by temperature, light and nutrients. The model enables spatial simulations by incorporating light extinction effects at the reactor scale (1 m) and nutrient absorption effects at the farm scale (1 km). Specifically, we simulated: 1. year-round productivities and N sequestration in the farm; 2. the farm size required for eutrophication mitigation in different seasons; and 3. spatial distribution of biomass production, chemical composition and environmental N along the farm in different dilution rates in the environment and in different airlift pumping flows. The simulations we presented refer to a theoretical estuarine environment comprising a constant 1D current, but was formulated so

that it could be later incorporated into complex oceanographic models (i.e. the $N_{env}$ equation follows the structure of the Convection–Diffusion equation, as used, for example, by the HAMOCC[53] and the CSIRO[54] models).

The high-resolution spatial and temporal model developed in this work, is an important step toward implementing precision agriculture techniques in seaweed aquaculture. Such advanced techniques are expected to improve productivities, efficiencies and accompanied environmental benefits, leading the way to sustainable marine development, accompanied by multiple economic and environmental benefits regarding climate change and water pollution mitigation.

Future studies need to validate the model on higher-resolution data of all state variables and engage in uncertainty quantification in different scales, including the farm-scale. In general, the robustness of the model will increase by further calibrating it with wider and more diverse empiric data sets, that will raise additional important constraining factors. Future efforts to improve the model should include adjusting it to P limited environments and relating to various phenomena that cause uncertainty in macroalgae cultivation. These phenomena include, for example, an unexplained decline in biomass, sudden sporulation, age, and history effect on the growth rate, water flow effects on growth and chemical composition and pest damage. By improving the ability to understand and describe both temporal and spatial phenomena in a seaweed farm in a resolution of days, these improved models should help to optimize the design of seaweed farms to combine environmental improvement and commercial viability.

## Methods

Our model incorporates multi-scale spatial effects: light extinction at the reactor scale and nutrient absorption at the farm scale, into a mathematical model of the *Ulva* sp. macroalgae metabolism[3] (See schematic description in Fig. 1). The spatial effects employ the following multiscale procedures: 1. from a single thallus scale (1 cm) to a reactor scale (1 m), relating to light extinction in the reactor, and 2. from a reactor scale to a farm-scale (1 km), relating to nutrient absorption in the farm. A Step-by-step formulation of the multi-scale model, starting at the thallus scale (Supplementary Figs. 2), is detailed in the Supplementary Methods.

The model was calibrated using experimental data from the reactor scale, relating to a 1.785 m³ U-shape bottom aerated (40–45 l min⁻¹) grazing proof cage reactor. Additional water (11.03 m³ per day) was pumped into the reactor from 1 m depth using four airlift pumps. *Ulva* sp. biomass was stocked in the reactor at a density of 1 kg FW m³ with an illuminated area of 2 m². Additional details about the reactor are available in Supplementary Figs. 3–4 and Supplementary Table 2

and in Chemodanov et al.[36]. After calibration the model was qualified with a sensitivity analysis.

Thereafter, biomass production rates, chemical compositions and farm-scale nitrogen removal was simulated under different seasons, levels of dilution in the environment (0-5% dilution ratio between every two reactors) and water-exchange rate in the reactor (0, 15, and 460 l h$^{-1}$). The detailed methodology of the work is presented in Fig. 7.

**Model assumptions**. The *Ulva* metabolic model assumes that the dynamics of the limiting nutrient, in this case nitrogen (N), under the constraining effects of environmental conditions (light intensity (I), temperature (T) and salinity (S)) predicates the dynamics of biomass growth and chemical composition. In the marine environment, the limiting nutrient is usually N[55] and our model focuses on N limited environments. However, similar models can be developed also for other elements such as phosphorus (P) and ferrous that may limit growth too in some marine environments. Our model also assumes that the organic carbon reserve, depending on carbon uptake and photosynthesis rates, is not limiting within the modeled conditions. The model follows the Droop Equation concept, in which the effect of the external, environmental, nutrient concentration on growth is mediated by internal nutrient concentrations ("cell quota")[18,56]. This is rather important as changes in internal N concentration occur gradually in a typical time scale of days whereas significant changes in environmental N concentrations may occur much faster, on a time scale of hours[57].

Our multi-scale model relates to cultivation in semi-closed reactors with controlled water exchange. This leads to the differentiation between nutrient concentrations inside the reactor that interact with the biomass directly, named here external N, and nutrient concentrations outside the reactor that are affected only secondarily, named here environmental N. Environmental N is the connecting agent that passes onwards in the flow the accumulating signal of changing N concentrations, which is translated into spatial differences in biomass composition and growth rate.

We used as a reference a cultivation reactor as described by Chemodanov et al.[36] (see above). Each reactor is assumed to be well-mixed by bottom aeration and is connected to airlift pumps that supplies the reactor with fresh seawater and nutrients. We also assume water flow through reactor boundaries is negligible.

We simulate the large-scale farm as composed of a continuum of macroscopic reactor size elements (compartments). This type of mass transfer model is commonly used in pharmaceutics which studies mass transfer through macroscopic units referred to as compartment[58]. The model assumes that the conditions in each reactor size control volume (compartment) can be accurately represented by one average value (external N) and that the domain of analysis (farm) is much larger than the macroscopic reactor size element.

We define our large-scale farm model as a 3D model (Supplementary Fig. 1). The x-axis is the direction of the flow and all simulations relate to one row of reactors in this direction. Each reactor constitutes an N sink, causing the spatial change of environmental N concentrations in the direction of the flow (x). By assuming the width of this change is small concerning the distance between the rows, this model becomes applicable also to multiple rows of reactors, with no variation in the y-axis. Finally, although light extinction increases with depth, potential variations in biomass with depth (z-axis) can be averaged out due to the well-mixed reactors' assumption.

**Model governing equations**. The multi-scale model is based on four governing ordinary differential equations (ODEs), describing the mass balance of four state variables: biomass density in a reactor (m, g Dry Weight (DW) l$^{-1}$, Eq. 1), biomass internal concentration of N ($N_{int}$, % gN gDW$^{-1}$, Eq. 2), external concentration of N in the reactor ($N_{ext}$, $\mu$mol $-$ N $l^{-1}$, Eq. 3) and the environmental N concentration outside the reactor ($N_{env}$, $\mu$mol $-$ N $l^{-1}$, Eq. 4) under varying temperatures, light intensities and salinities.

$$\frac{\partial m}{\partial t} = (\mu - \lambda)m,$$

$$\mu = \mu_{max}f, \quad f = f_{Temp}f_S \min\{f_{N_{int}}, f_{P_{int}}, f_I\}$$

$$\text{Initial Condition(I.C)}: m_{(x,t=0)} = m_0 \tag{1}$$

Where $\mu_{max}$ (h$^{-1}$) is the maximum specific growth rate and $f$ is the combined growth function, made of $f_{Temp}, f_s, f_{N_{int}}, f_{P_{int}}$ and $f_I$, which are the T, S, $N_{int}$ $P_{int}$ and I growth functions[3] (see full equations in Supplementary Methods). This function includes also light as a potential limiting factor under Leibig's law of the minimum, regardless of the difference between light and nutrient growth mechanisms, as appears in previous works[42,59]. $\lambda$ is biomass specific losses rate as a function of T and is formulated of $\lambda_{20}$ (h$^{-1}$), the specific rate of biomass losses and $\theta$, an empiric factor of biomass losses[3]. $\lambda$ does not include losses by grazing, sporulation and fragmentation by storms, which vary between different environments and are highly affected by extreme events. We adjusted daily specific growth and losses rates to hourly rates, assuming for simplicity that growth and biomass losses occur only during light hours (see details in Supplementary Methods). This assumption ignores night growth that occurs due to metabolites produced during light-time photosynthesis[60], and thus distorts growth distribution throughout the day.

However, the assumption does not affect total daily growth and therefore does not impair the model accuracy at a temporal resolution of days to weeks.

$$\frac{\partial N_{int}}{\partial t} = \psi_{N_{ext}} - N_{int}fm$$

$$\psi_{N_{ext}} = \frac{N_{intmax} - N_{int}}{N_{intmax} - N_{intmin}} \frac{V_{max}N_{ext}}{K_S + N_{ext}}$$

$$\text{I.C}: N_{int(x,t=0)} = N_{int0} \tag{2}$$

Where $\psi_{N_{ext}}$ ($\mu$mol-N gDW$^{-1}$ h$^{-1}$) is the N uptake function, formulated of $N_{intmax}$ and $N_{intmin}$ (%gN gDW$^{-1}$), the maximum and minimum $N_{int}$ concentrations, respectively, $V_{max}$ ($\mu$mol-N gDW$^{-1}$ h$^{-1}$), the maximum N uptake rate and $K_S$ ($\mu$mol-N l$^{-1}$), the N half-saturation uptake constant. $-N_{int}fm$ describes $N_{int}$ dilution in biomass by growth.

$$\frac{\partial N_{ext}}{\partial t} = \frac{Q_p(N_{env} - N_{ext})}{V_{cage}} - \psi_{N_{ext}}m$$

$$\text{I.C}: N_{ext(x,t=0)} = N_{ext0} \tag{3}$$

Where $Q_p$ (l h$^{-1}$) is the airlift pumping flow and $V_{cage}$ (m$^3$) is the reactor volume. The change in $N_{ext}$ is the sum of N in incoming airlift pump flow, N in reactor overflow and N uptake by the biomass in the reactor.

$$\frac{\partial N_{env}}{\partial t} = \frac{[-Q_s(N_{env_{x-1}}(1-d) - N_{env_x}) - Q_p(N_{env_x} - N_{ext_x})]}{V_{cage}}$$

$$\text{I.C}: N_{env(x,t=0)} = N_{ext0}, \text{ Boundary Condition(B.C)}: N_{env(x=0,t)}N_{env(x=0,t)} = N_{ext0} \tag{4}$$

Where $N_{env_x}$ is $N_{env}$ below reactor x at time t, $N_{env_{x-1}}$ is $N_{env}$ below reactor x-1 at time t, $d$ (%) is the dilution ratio between every two reactors and $Q_s$ (l h$^{-1}$) is the stream flow through an area equivalent to the reactor narrow-side cross-section. Thus, the change in $N_{env}$ is the sum of incoming N flows (upstream flow and reactor overflow) and outflowing flows (downstream flow and airlift pumping into the reactor). This form of Convection-Diffusion equation may be adjusted in the future to fit also more complex hydrodynamic models (i.e. dynamic 2/3D currents compared to a constant 1D current simulated in this work). All four ODEs were solved numerically with hourly time steps.

**Scale elements in model**. The multi-scale model has two scale elements: 1. light extinction at the reactor scale that requires dynamic averaging of light intensity per biomass unit, and 2. nutrient absorption at the farm scale that requires following the dynamics of environmental N.

*Single thallus to reactor*. In the metabolic model of a single thallus scale, growth is affected directly by incident light intensity (Eq. 5). This function follows the commonly used Monod model but could be replaced also by alternative form, such as the hyperbolic tangent model or the simplified light-inhibition model, all acknowledged in the literature[42]. In transition to a reactor scale, light intensity is averaged per biomass unit, as formulated by Oca et al.[11] (Eq. 6). This formulation considers water depth in the reactor, biomass density and light extinction coefficients of both water and biomass. In both equations, we multiplied $I_0$ by a 0.43 PAR constant, representing the ratio of the sunlight which is suitable for photosynthesis[61].

$$f(I) = \frac{I}{K_I + I}PAR \tag{5}$$

Where $I$ and $K_I$ ($\mu$mol photons m$^{-2}$ s$^{-1}$) are incident light intensity and light half-saturation constant, respectively.

$$f(I) = \frac{I_{average}}{K_I + I_{average}}PAR$$

$$I_{average} = \frac{I_0}{K_0 Z + K_a SD}[1 - \exp(-(K_0 Z + K_a SD))] \tag{6}$$

Where $I_{average}$ and $I_0$ ($\mu$mol photons m$^{-2}$ s$^{-1}$) are average photon irradiance in the reactor and incident photon irradiance at water surface, respectively, $SD$ (gDW m$^{-2}$) is stocking density of biomass per unit of water surface in the reactor, $K_0$ (m$^{-1}$) is water light extinction coefficient, $Z$ (m) is maximum water depth in the reactor and $K_a$ (m$^2$ gDW$^{-1}$) is *Ulva* light extinction coefficient.

*Reactor to farm*. In a single well-mixed reactor, nutrient reduction by biomass is local and does not accumulate along the stream. Therefore, Eq. 4, describing changes in $N_{env}$, is redundant. However, in a seaweed farm, spatial variations in $N_{env}$ cannot be described without Eq. 4 that connects the reactors and the environment. Equation 3, describing changes in $N_{ext}$, was derived from the Convection–Diffusion equation[62] (Eq. 7). Equation 4, describing changes in $N_{env}$, is

based on the same equation, without the uptake term.

$$\frac{\partial N_{ext}}{\partial t} = \nabla \cdot (D \nabla N_{ext}) - \nabla \cdot (v N_{ext}) - \psi_{N_{ext}} m \quad (7)$$

Where $D$ (m$^2$ s$^{-1}$) is the average diffusivity coefficient of dissolved inorganic N species and v (m s$^{-1}$) is the velocity field in which the dissolved nitrogen is moving. Both Eq. 3 and Eq. 4 are derived from this equation, with specific simplifying assumptions: 1. $D$ is constant in space; 2. incompressible velocity flow, and 3. zero net diffusivity, as the reactor is well-mixed and there is no concentration gradient ($\nabla N_{ext} = 0$). Therefore, $N_{ext}$ in the reactor is affected only by the N supply by airlift pump (normalized to reactor volume) and N uptake by algae. Equation 4, describing changes in $N_{env}$, follows the same principal form but without the uptake term.

**Model calibration**. We calibrated the model parameters using experimental growth data of *Ulva* cultivation in a single well-mixed sea-based reactor from Chemodanov et al.[36] (Supplementary Figures 3-4 and Supplementary Table 2). First, we determined the *Ulva* light extinction coefficient, $K_a$, by minimizing the root mean relative error (RMSR$E_1$, Eq. 8) between modeled biomass growth from three experiments based on: 1. in situ measured light intensity (Onset HOBO Pendant®), and 2. light intensity data extracted from the IMS data base from the Israel Meteorological Services (https://ims.data.gov.il/he/ims/6). This was done by calculating RMSR$E_1$ for 20 values and 320 different parametric combinations of $\mu_{max}$, $K_a$, $K_1$, $n$, $T_{max}$, $T_{opt}$, and $\lambda_{20}$ in a defined range and identifying the $K_a$ value which results in the minimal errors. One experiment, where biomass degradation could not be explained by the model, was omitted from the calibration process. Next, using the same method, we determined the values of $\mu_{max}$, $K_I$, $n$, $T_{max}$, $T_{opt}$ and $\lambda_{20}$ (20 values and 280 different parametric combinations) by minimizing the mean error between measured and modeled biomass growth (N = 4) using in situ temperature data when available (in 3 out of 4 experiments) or IMS data otherwise, and IMS light intensity data (RMSR$E_2$, Eq. 9).

$$RMSRE_1 = \sqrt{\frac{\sum_{i=1}^{N} \left| \frac{PV_{mIn} - PV_{mEx}}{PV_{mIn}} \right|}{N}}, \ N = 3 \quad (8)$$

$$RMSRE_2 = \sqrt{\frac{\sum_{i=1}^{N} \left| \frac{m_f - PV_{mEx}}{m_f} \right|}{N}}, \ N = 4 \quad (9)$$

Where RMSR$E_1$ is the Root Mean Square Relative Error between $PV_{mIn}$ (g DW l$^{-1}$), the predicted value of final biomass based on in-situ light intensity data and $PV_{mEx}$ (g DW l$^{-1}$), the predicted value of final biomass based on ex-situ light intensity data, and RMSR$E_2$ is the Root Mean Square Relative Error between $m_f$ (g DW l$^{-1}$), the measured final biomass and $PV_{mEx}$.

Data from returns 2, 4, and 5 (Supplementary Table 2) was used for the calibration of the light extinction coefficient, $K_a$, and data from returns 1, 2, 4, and 5 was used for the calibration of $\mu$, $K_a$, $K_1$, $n$, $T_{max}$ and $T_{opt}$. Return 3 was not used for calibration as its negative growth could not be explained by the model.

**Sensitivity analysis**. To examine how each parameter, in a defined range (Supplementary Table 4), influences model simulations output, we analyzed farm-scale sensitivity of state variables using SALib, the Sensitivity Analysis Library in Python[63]. Specifically, the analysis focused on the projected values of total produced biomass, total accumulated $N_{int}$ and average final $N_{env}$, under the simulation frame of a 100-reactors' farm and a cultivation period per season, that should suffice to observe both temporal and spatial effects of the different parameters. First, 10 values and 420 random parametric combinations of all model parameters (Supplementary Table 4) were generated using the Saltelli method[64,65]. Next, each combination was run through the model, producing an array of possible biomass production, N accumulation and final $N_{env}$ results. Finally, the results were analyzed using the Sobol analysis[66], giving each parameter a first order and total sensitivity index between zero and one.

**Model simulations**. The model was applied to simulate year-round cultivation of *Ulva* sp. in a row of cultivation reactors in a nutrient-enriched estuary environment located in a semi-arid climate. Data regarding nutrient concentrations, salinities, water temperature and flow was taken from the long-term study of Suari et al.[67] on the Alexander estuary, located in the center of Israel (Supplementary Fig. 5 and Supplementary Table 3). I data was extracted from the IMS database from the Israel Meteorological Services (https://ims.data.gov.il/he/ims/6). Although S varies with depth and can change dramatically according to flesh flood events and formation of sandbar breaches[67], effect on growth was minor and we used a constant value of S=30 PSU. All constraining environmental factors except nutrients were assumed to be constant in space. Each cultivation cycle started with a constant set of initial conditions ($m_0$, $N_{int,0}$, $N_{ext,0}$ and $N_{env,0}$) which applied to all reactors. Harvesting back to initial biomass was performed every two weeks, and accumulated biomass production was calculated. In addition, N removal from the environment was calculated as the difference between total N in final and initial biomass. Specific simulations of seasonal N removal capacity were used to project

the number of reactors needed to achieve a 10 μM-N level threshold, which is below levels found in extremely eutrophicated zones[48], in each season. Finally, a spatial perspective was added by examining the system dynamics under various pumping levels and in a diluting environment, in which the enriched $N_{env}$ water is diluted by mixing with lower $N_{env}$ water (i.e. 5% dilution between each two reactors).

## Statistics and reproducibility

The model developed in this study was calibrated versus growth results from four independent cultivation experiments which were reported in Chemodanov et al.[36].

**Reporting summary**. Further information on research design is available in the Nature Research Reporting Summary linked to this article.

## Data availability

The authors declare that all data supporting this study, specifically for model calibration and simulations, is available within the paper and its Supplementary Information files. Experimental results used for calibration are presented in Supplementary Table 2. Temperature and light intensity data are available in https://doi.org/10.5281/zenodo.4062432 [68]. Other environmental data is available in Supplementary Table 3.

## Code availability

The entire code of this study, written it Python (3.7.3), is available as an open source in https://doi.org/10.5281/zenodo.4062432 [68].

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

## Acknowledgements

The authors thank the TAU-Berkeley fund for funding and Tzachi Yaffe for assistance in the graphic design of figures.

## Author contributions

M.Z. has designed and performed the research, analyzed the data, and wrote the manuscript. B.R., A.L. and A.G. have supervised, contributed analytic tools, and revised the manuscript.

## Competing interests

The authors declare no competing interests.
