## [Peer Review File · Communications Biology]

Reviewers' comments:

Reviewer #1 (Remarks to the Author):

This paper reported a model for macroalgae growth and nitrogen sequestration in an intensive cultivation farm. The macroalgae cultivation system here seems interesting and could be considered for future implementation. However, for the model itself, although it was well constructed, and seems that this model can be used to predict performance of the macroalgae cultivation system, I don't think it was novel enough to be accepted by CB. The model consists of some ordinary equations which were all well studied before, and the results of prediction were quite normal, so I don't think this paper has significant theoretical contributions to biology or ecology. I suggested the author resubmitted it to another journal.

Here are some suggestions:

1. For the equation (1), I doubted that the "I (light)" can be included in $\min\{ \}$, as the mechanism for "light-growth" is different from N and P. I don't think Monod equation was the right one for describing light-growth interaction. There are some kinds of "light-primary productivity" model or "light-limit growth" model. You can refer to them.
2. It seems that the macroalgae cultivation system was expected to install in the field. I don't think the flow direction can be well controlled precisely, and the wind, wave and even tide would also affect your system. Therefore, the situation would be far from the theoretical scenario.
3. It is better to present some results of larger-scale field experiment to prove the feasibility of the model, while you only use the results of several cages to provide parameters for the model.

Reviewer #2 (Remarks to the Author):

Communications Biology

Manuscript Number: COMMSBIO-20-3005

The manuscript titled "Multi-Scale Modeling of Intensive Macroalgae Cultivation and Marine Nitrogen Sequestration" is interesting and can be published in Communications Biology. However, few minor suggestions are provided for the improvement.

-
1. The research gap related to the limited models and shortcomings of the existing models in predicting the biomass yield, nitrogen composition and biochemical composition may clearly be stated at the end of the Introduction.
 2. The detailed methodology of the work may be presented in a simple Flowchart. Provide the technical details of the small scale reactor/farm adopted in the study.
 3. The literature review may include previous works on a similar topic. The following may be useful: <https://doi.org/10.1016/j.rser.2018.03.100>
 3. Please critically analyze and discuss the usefulness and limitations of the proposed theoretical "Multi-scale macroalgae growth models" in the discussion section.
 4. Accuracy of the model and its validation of the results may be compared with prior published literature.
 5. Some of the keywords may be updated. A comprehensive list compiled from the work is enclosed for the authors' consideration.
Suggested: marine sustainable development; temperature, light and nutrients; Biosequestration; Bioremediation; Biorefinery Feedstock; macroalgae growth; nitrogen sequestration; Biomass production rates, chemical compositions; macroalgae-based bioeconomy; seaweed aquaculture
 6. What are the significant contributions of work towards the global community and audience towards future deployment of macroalgae farms?
 7. Figure 3 may include environmental N. Sequestration
 8. It would be very interesting to see the capability of the model to predict the biochemical composition of the species throughout the year.
 9. Please justify the variation of the biomass yield and N. Sequestration (Figure 3). Peak in March and low in August.
 10. N. Sequestration and biomass yield almost follow the same trend. Obtain the correlation equation between them.

Reviewer #1

This paper reported a model for macroalgae growth and nitrogen sequestration in an intensive cultivation farm. The macroalgae cultivation system here seems interesting and could be considered for future implementation. However, for the model itself, although it was well constructed, and seems that this model can be used to predict performance of the macroalgae cultivation system, I don't think it was novel enough to be accepted by CB. The model consists some ordinary equations which were all been well studied before, and the results of prediction were quite normal, so I don't think this paper has significant theoretical contributions to biology or ecology. I suggested the author resubmitted it to another journal.

Response: We thank the reviewer for finding our paper interesting and the model well-constructed and for the suggestions used to improve our paper. We acknowledge the fact that the model is based on ordinary equations which are quite normal. However, we believe our paper does have significant theoretical contributions to macroalgae farming and nutrient biosequestration, which are two important fields that are based on biological understanding and technologies. The model is novel as it advances the modeling abilities of the biological components of macroalgae farming, **adding spatial components and enabling the analysis of large-scale activities based on reactor scale experimental data**. We are not aware of including these important terms in previously published literature.

Here are some suggestions:

1. For the equation (1), I doubted that the "I (light)" can be included in $\min\{ \}$, as the mechanism for "light-growth" is different from N and P.

Response: We thank the reviewer for this comment. We used this form of μ (growth rate function) following the Leibig's law of the minimum, and specifically the "Type I. Colimitation - minimum form" presented in equation 3 in the work by Saito et al. (2008):

<https://aslopubs.onlinelibrary.wiley.com/doi/pdf/10.4319/lo.2008.53.1.0276>.

The same work relates also to light as a "nutrient" or "substrate" in the context of limiting factors, and we adopt this definition as it fits our understanding of the algae growth, which is usually limited by light (i.e. leading to proteins accumulation) or by nitrogen (i.e. leading to protein dilution and carbohydrates accumulation). We added a clarification in the text in rows 363-364. In this context see also

<https://link.springer.com/article/10.1007/s12155-019-10036-3> and

https://www.sciencedirect.com/science/article/pii/S0734975013001481?casa_token=JoT6qlfeRqYAAAAA:E1Kalu68JI0zq36wZz1LuEK-EPs6RZ_6dcO4bliqZQVkJekThYmDBGyUrhWi8_WgPMPieL86Zg#t0005

2. I don't think Monod equation was the right one for describing light-growth interaction. There are some kinds of "light-primary productivity" model or "light-limit growth" model. You can refer to them.

Response: We thank the reviewer for this important comment. As mentioned in the work of Béchet et al. (2013), different light-growth interaction equations exist. Some authors recommend using the hyperbolic tangent model, some recommend using the simplified light-inhibition model and also the Monod form models are very common. We chose the Monod form, which was used also by the author from which we adopted the light absorption function ($I_{average}$, eq 6) - Oca et al. (2019):

<https://link.springer.com/article/10.1007/s10811-019-01767-z>. In any case, the general behavior of the model is not expected to change significantly behave if a different form of light-growth function will be used in the future.

3. It seems that the macroalgae cultivation system was expected to install in the field. I don't the flow direction can be well controlled precisely, and the wind, wave and even tide would also affect your system. Therefore, the situation would be far from the theoretical scenario.

Response: We thank the reviewer for this comment. In this stage we applied the model only on a simple theoretical scenario. However, looking towards the future, we formulated the model based on the convection-diffusion equation, as used by complex oceanographic model, so that in later stages it could be incorporated into such an advanced model such as HAMOCC (<https://agupubs.onlinelibrary.wiley.com/doi/full/10.1029/2012MS000178>) and CSIRO (<https://gmd.copernicus.org/articles/13/4503/2020/>). We related to this issue in rows 279-283 in the "Results and Discussion" chapter and in rows 385-387 in the "Methods" chapter.

4. It is better to present some results of larger-scale field experiment to prove the feasibility of the model, while you only use the results of several cages to provide parameters for the model.

Response: We thank the reviewer for this comment. We agree that it is better to present results from large-scale field experiments, however, such experiments are out of the scope of this work. Although the results are based on field work in the reactor scale, in this stage the model remains theoretical for the farm-scale. This is mentioned as a limitation of this work and pointed out as a gap for future studies. See rows 262-266 and 289-290.

Reviewer #2

Communications Biology

Manuscript Number: COMMSBIO-20-3005

The manuscript Titled " Multi-Scale Modeling of Intensive Macroalgae Cultivation and Marine 3 Nitrogen Sequestration" is interesting and can be published in Communications biology. However few minor suggestions are provided for the improvement

Response: We thank the reviewer for this finding our paper interesting and for his valuable suggestions, used to improve the paper.

1. The research gap related to the limited models and shortcoming of the existing models in predicting the biomass yield, nitrogen composition and biochemical composition may clearly be stated at the end of the Introduction.

Response: We thank the reviewer for this comment. As suggested, we added a clear statement regarding the research gap at the beginning of the last paragraph of the introduction. See rows 75-77.

2. The detailed methodology of the work may be presented in a simple Flowchart.

Response: We thank the reviewer for this comment. As recommended, we added a simple flow chart representing the detailed methodology. See Fig. 2 and row 317.

3. Provide the technical details of the small scale reactor/farm adopted in the study

Response: We thank the reviewer for this comment. As recommended, we added the technical detail of the reactor scale adopted to this study. See rows 308-312.

4. The literature review may include previous works on a similar topic. The following may be useful.

<https://doi.org/10.1016/j.rser.2018.03.100>

Response: We thank the reviewer for this comment. We rephrased a bit and referred to additional previous works on macroalgae biorefineries, including the recommended article. See rows 32-35.

5. Please critically analyze and discuss the usefulness and limitations of the proposed theoretical “Multi-scale macroalgae growth models” in the discussion section

Response: We thank the reviewer for this comment. As recommended, we added a paragraph analyzing and discussing the limitations of the proposed model in the discussion chapter. See rows 262-269. The usefulness of the model is mentioned in the discussion and in the conclusions. See rows 248-257 and 280-284.

6. Accuracy of the model and its Validation of the results may be compared with prior published literature.

Response: We thank the reviewer for this important comment. We now compared the accuracy of the model to other biological model in literature and demonstrated its advancement compared to similar models previously published.

7. Some of the keywords may be updated. A comprehensive list compiled from the work is enclosed for the authors consideration.

Suggested: marine sustainable development; temperature, light and nutrients; Biosequestration; Bioremediation; Biorefinery Feedstock; macroalgae growth; nitrogen sequestration; Biomass production rates, chemical compositions; macroalgae-based bioeconomy; seaweed aquaculture

Response: We thank the reviewer for this comment. As recommended, we have updated some of the keywords (lines 12-13), and added the following keywords:

Marine Sustainable Development
Macroalgae-Based Bioeconomy

8. What are the significant contribution of work towards the global community and audience towards future deployment of macroalgae farms

Response: We thank the reviewer for this comment. The whole final paragraph of the “results and discussion” chapter relates to the significant contribution of the work. In addition, we added a concluding sentence, emphasizing the contribution of the work towards future deployment of macroalgae farms. See rows 257-261.

9. Figure 3 may include environmental N Sequestration

Response: Figure 3 (updated to Fig. 4) already includes environmental N sequestration. We would like to refer the reviewer to the blue stars representing the environmental N sequestration.

10. It would be very interesting to see the capability of the model to predict the biochemical composition of the species throughout out the year

Response: We thank the reviewer of this comment. The capability of the model to predict the biochemical composition throughout the year is demonstrated in Fig. 5c. We related to this feature in rows 180-182.

11. Please justify the variation of the biomass yield and N Sequestration (Figure 3). Peak in March and low in August

Response: We thank the reviewer for this comment. Seasonal variations in yields and N sequestration results largely from variations in water temperatures - optimal T leads to high yield and high T leads to very low yields. See rows 156-160.

12. N Sequestration and biomass yield almost follow the same trend. Obtain the correlation equation between them.

Response: We thank the reviewer for this comment. We added the Pearson's r values that show the very high correlation between biomass production and N sequestration, and added a comment that in larger farms or in diluting environments, in which environmental N could be depleted, this correlation is expected to decrease. See rows 149-155.

REVIEWERS' COMMENTS:

Reviewer #2 (Remarks to the Author):

The revised manuscript titled "Multi-Scale Modeling of Intensive Macroalgae Cultivation and Marine Nitrogen Sequestration" is much improved in terms of the presentation and clarity